# Screening for prevalence of current TB disease and latent TB infection in type 2 diabetes mellitus patients attending a diabetic clinic in an Indian tertiary care hospital

Pradipkumar Arvindbhai Dabhi[1], Balamugesh Thangakunam[1], Richa Gupta[1], Prince James[2], Nihal Thomas[3], Dukhabandhu Naik[4], Devasahayam Jesudas Christopher[1] *

1 Department of Pulmonary Medicine, Christian Medical College, Vellore, Tamil Nadu, India, 2 Department of Respiratory Medicine, Christian Medical College, Vellore, Tamil Nadu, India, 3 Department of Endocrinology, Diabetes and Metabolism, Christian Medical College, Vellore, India, 4 Department of Endocrinology, Jawaharlal Institute of postgraduate medical education and Research (JIPMER), Pondicherry, India

* djchris@cmcvellore.ac.in

**Data Availability Statement:** All relevant data are available from the Harvard Dataverse (DOI: 10. 7910/DVN/H0TY7Y).

## Abstract

### Background

Diabetes triples the risk of developing tuberculosis (TB). This study was designed to determine the prevalence of past and current TB disease and Latent TB infection (LTBI) in type 2 Diabetes Mellitus (NIDDM) patients.

### Design

This was a prospective descriptive study on all NIDDM patients attending a Diabetic clinic. Detailed history, included details of previous history of TB (Past TB)and symptoms of active TB and a thorough physical exam was also done. When clinical suspicion of TB was present, appropriate investigations were carried out to diagnose 'Current TB'. Subsequently, 200 consecutive patients who were negative for Past and Current TB were screened for Latent TB infection (LTBI) by tuberculin skin test.

### Results

Of 1000 NIDDM patients enrolled, 43(4.3%) had Past TB. Of remaining 957 patients, 50 were evaluated for New TB on the basis of suggestive symptoms and 10(1%) patients were confirmed to have Current TB. Risk factors for Past or Current TB 'DM-TB' in comparison with 'DM Only' group were; male sex (72% VS 57%; P = 0.033), manual laborer (28% VS 15%; P = 0.012), smoking (26% VS 14%; P = 0.015), alcohol consumption (23% VS 9%; P<0.001)& being on treatment with Insulin (40% VS 20%; P<0.001). There was a protective effect with being a home maker (17% VS 37%; P = 0.034&overweightstatus (53% VS 71%; P = 0.004). Of the 200 patient without Past or Current TB, who were screened for LTBI, 96

**Funding:** This study was partly supported by the Department of Biotechnology (India) and National Institute for Allergy and Infectious Diseases & National Institutes of Health (USA). The support was administered by CRDF Global under the aegis of the RePORT India consortium. We also acknowledge part funding from Fluid Research grant CMC, Vellore.

**Competing interests:** The authors have declared that no competing interests exist.

(48%) patients were found to have LTBI. Male sex was the only significant risk factor for LTBI (72% VS 59%; P = 0.05).

## Conclusion

Past and Current TB was substantial in patients attending a Diabetic Clinic. Active symptom screening for TB in these clinics could lead to increase in case detection and earlier diagnosis.

## Background

While significant gains in global TB control have been achieved, an estimated 10 million people developed TB disease and 1.2 million HIV negative patients died from the disease in 2018 [1]. India is the highest TB burden country accounting for an incidence of 2.69 million cases. As per India TB report 2017 India accounts for a quarter of world's TB cases and Government of India plans to achieve a rapid decline in burden of TB, morbidity and mortality while working towards elimination of TB in India by 2025 [2]. Type 2 diabetes mellitus (DM) triples the risk of developing tuberculosis (TB), and rates of TB are higher in people with DM than in the general population [3]. The link of DM and TB is important in developing countries where TB is endemic and the prevalence of DM is rising. The prevalence of diabetes in adults aged 20 years or older in India increased from 5·5% in 1990 to 7·7% in 2016 [4] Although the definite pathophysiological mechanism of the effect of DM as a risk factor for TB is unknown, some hypotheses are: depressed cellular immunity, dysfunction of alveolar macrophages, low levels of interferon gamma, pulmonary microangiopathy, and micronutrient deficiency [5,6].The coexistence of these TB and DM in low and middle income country is an example of a bidirectional association between a communicable and non-communicable disease that increases the dual burden of both the diseases [7].

Some believe that screening for active TB among diabetics could improve case detection, earlier treatment and prevent transmission of disease [8].The World Health Organization (WHO) and the international union against lung and tuberculosis disease (IUALTD) launched the 'Collaborative framework for the Care and Control of Diabetes and Tuberculosis', with one important recommendation being the routine implementation of bi-directional screening of the two diseases [9]. While studies in India have show that bidirectional screening for DM and TB is feasible [10]. High yield of DM has been reported among TB patients but the yield of TB among DM patients was low and so further research using new, improved TB diagnostic tools was recommended [11].Other low income countries have reported lower prevalence of DM among patients with TB [12].

In the present study, we aimed to determine the prevalence of Past and Current TB disease and Latent TB infection (LTBI)in patients with NIDDM attending a Diabetic clinic and to describe the prevalence, socio-demographic characteristics, clinical features, microbiological and biochemical variables of these patients.

## Study design

A prospective screening of the DM patients attending the Diabetic clinic of the department of Endocrinology and Diabetes for Past and Current TB disease and LTBI, between September 2014 to April 2015.

## Setting

The study was conducted at the Christian Medical College and Hospital, Vellore, a tertiary care referral hospital in India.

## Inclusion criteria

Patients with NIDDM18 years or older.

## Exclusion criteria

1. Acute medical illness

2. Known HIV or other immunosuppressive disease

3. Unable or unwilling to comply with study protocol

4. Pregnancy

## Methods

Consecutive NIDDM patients, attending the Diabetic clinic of the Christian Medical College, Vellore, a referral teaching hospital in Southern India, who consented to participate in the trial and provide informed consent were recruited. A detailed history was obtained and in particular past history TB, details of organ involvement and treatment details.

In addition, information about socio-demographic characteristics, history of smoking and alcohol consumption and duration of DM and current medication for DM were obtained. Weight, height, BMI and the most recent blood glucose measurements (fasting & post prandial) and glycosylated hemoglobin (HBA1c) results were recorded. BMI of greater than 23 was considered as high risk for DM in our population [13]. For patients with no chest radiographs within past 1 year, a fresh chest X-Ray (PA view) was obtained. Both these were reviewed by experienced pulmonary physicians and the radiological features were recorded

They fell in one of these 4 categories:

1. Patients with past history of TB (Past TB)

2. Patients with features of–pulmonary or extra-pulmonary TB (TB suspects)

3. TB suspects who were proven to have TB (Current TB)

4. Patient with no past history or current clinical features of pulmonary or extra-pulmonary TB (Non TB)

The patients were evaluated as follows:

1. Past TB: Details of organ involvement, history, mode of diagnosis and treatment details were recorded. For those with history of pulmonary TB (PTB), old chest radiographs, if available were reviewed

2. TB suspects: The patients with current symptoms of active TB with the following symptoms:

   - Productive cough for >2 weeks, fever, night sweats, loss of appetite and weight loss.

   - Symptoms and signs of extra-pulmonary TB like lymphadenopathy, pleural effusion, typical skin lesion and joint swelling/pain etc.

- These patients were further investigated to confirm active TB disease. Chest x-ray and 3sputum examinations (2 for AFB smear and 1 for X-pert MTB/Rif and MGIT culture) were done for suspected active PTB patients. Sputum for AFB smear was done by Ziehl-Neelsen staining method, X-pert MTB/Rif & MGIT cultures were performed as per the manufacturer's instruction. For extra-pulmonary TB (EPTB), appropriate aspirate or tissue was obtained for histopathological evaluation and MGIT culture.

3. Those subsequently diagnosed with active TB (Current TB) were initiated on standard anti TB treatment.

4. Among these patients with no Past or Current TB (Non TB), 200 consecutive patients were tested for LTBI using one step tuberculin skin test (TST). TST was performed by using tuberculin 2TUsolution (PPD RT 23). TST was considered as positive if the indurations were ≥ 10mm and these were categorized as having 'LTBI'. Those who underwent TST test and were negative (≤ 10 mm) and those without LTBI were categorized as 'No LTBI',

5. Those with Past and Current were grouped as 'DM-TB' the rest of the patients were grouped as 'DM Only'

The study subjects flowchart is shown in Fig 1. All patient data was recorded on case report form and captured in an electronic data base.

## Statistical analysis

Analysis and statistics Data were extracted from the electronic database and analyzed using SPSS (Statistical Package and Service Solutions, version 16.0, SPSS Inc, Chicago, IL). The socio-demographic characteristics were tabulated. Mean and standard deviations (SD) were calculated. Continuous variables such as age, BMI and duration of DM were converted to categorical variables and compared using the $\chi^2$ test where appropriate. Levels of significance were set at 5%.The clinical features, diabetic control variables and diabetic treatment details of patients with 'DM-TB' were compared with the 'DM Only' group and the 'LTBI 'group was compared with 'No LTBI' group.

## Ethics approval

Ethics approval for the study was obtained from the Institutional review board of the Christian Medical College Vellore.

## Results

Of 1000 DM patients screened for TB, the mean age was 51 [SD 11.3] years, 586 [59%] were male and 414 [41%] were female). In all, 43(4.3%) hadPastTB, and none of these patients had features of active TB (relapse) at screening. Among the other 957 patients, there were 50 patients with clinical suspicion of TB, who were investigated further to confirm/rule out TB. Ten (20%) of those investigated turned out to have Current TB. In all, there were 53 cases of Past and Current TB 'DM-TB'.

Table 1 shows the details of the 10 patients diagnosed to have Current TB-9 of them had pulmonary TB(PTB) and 1 extra pulmonary TB. Out of the 9 pulmonary TB patients, 8 were positive by smear as well as X-pert MTB/Rif, while 1 patient was negative by both these test but had MTB growth on MGIT culture. Out of 9 PTB patients, 8 were drug sensitive and 1 drug resistant (MDR) TB. The only patient with extra-pulmonary TB had cervical lymphadenopathy, confirmed by biopsy showing caseating granuloma on histopathology. MTB did not grow on culture, the patient responded to anti-TB treatment.

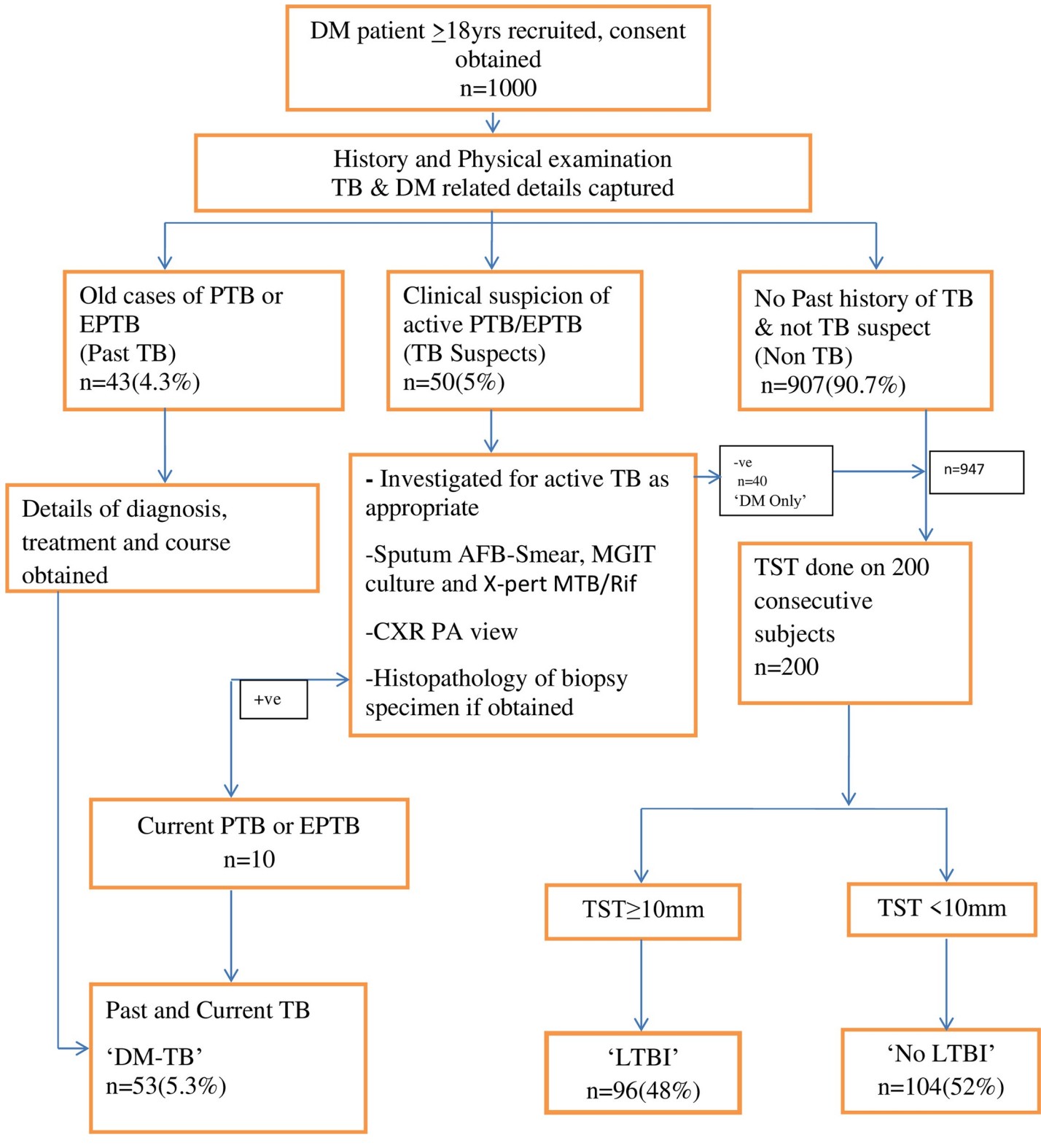

**Fig 1. Study subjects flowchart.**

**Table 1. Types and categories of active TB in DM patients screened for TB in a tertiary care hospital between September 2014 to April 2015.**

| Newly diagnosed with TB–'Current TB' (n-10) | |
|---|---|
| Category and type of TB | |
| Pulmonary TB | 9 |
| AFB Smear positive | 8 |
| X-pert MTB Rif positive (Rif sensitive-07,Rif resistance-01) | 8 |
| Smear and X-pert MTB Rif negative, Culture positive(drug sensitive) | 1 |
| Extra pulmonary TB | 1 |

The socio-demographic 'DM-TB' and 'DM Only' patients are shown in Table 2.Risk factors for Past or Current TB 'DM-TB' in comparison with 'DM Only' group were; male sex(72% VS 57%; P = 0.033), manual laborer (28% VS 15%; P = 0.012), smoking(26% VS 14%; P = 0.015), alcohol consumption(23% VS 9%; P<0.001), I & being on treatment with Insulin(40% VS 20%; P<0.001). There was a protective effect with being a home maker (17% VS 37%; P = 0.034)& overweight status (53% VS 71%; P = 0.004).

The clinical variables and DM treatment are compared between 'DM-TB' and 'DM Only' patients (Table 2). Those with normal BMI was significantly lower in the 'DM-TB' group (52.8%VS 71.2%), conversely those with high BMI were higher in the 'DM Only' group. With regards to treatment, significantly less patients in the 'DM-TB' group were on oral hypoglycemic agents (OHAs) only (26% VS 51%) and significantly more were on insulin (40% VS 20%). There was no difference between the groups with regards to HbA1c, fasting and post prandial plasma glucose.

LTBI was positive in 96 of the 200 patients tested (48%). The 'LTBI' group was compared with the 'No LTBI' group (Table 3). A higher proportion of 'LTBI' patients were male (72% VS 59%; P = 0.05). There was no other difference between the groups.

## Discussion

While there are studies that have addressed the prevalence of DM in TB clinics, prevalence of TB among Diabetics are fewer and to the best of our knowledge, there are no studies from our country on LTBI prevalence among NIDDM patients. There are few studies on LTBI in the general population and these reported a higher prevalence of LTBI among patients with DM, with the rates ranging from 28.2 to 42.4%. [14, 15]

While bi-directional screening has been shown to be feasible, A Majumdar et al. recently reported that the implementation of bidirectional screening was poor in the public health system. Adequate staffing, regular training, continuous laboratory supplies for DM diagnosis and widespread publicity were recommended [11].

In all, there were a total of 53 (5.3%) who either had Past or Current TB among patients attending our Diabetic clinic. This is substantial. Among the 10(1%) patients with current TB, 8(80%)had smear-positive disease and were therefore infectious. The prevalence of active TB in this population works out to 1000 per 100000 patients, which is higher than the total TB case notification rate in India. [16]

S. Kumpatlaetal [17], screened 7083 patient of DM for TB and found that only 50(0.7%) patients had current or Past TB, which was substantially lower than what we found (5.3%). Another study by Mave et al. which screened 630 patients in a public sector tertiary care hospital failed to yield any active TB case, using a WHO-recommended questionnaire among people with DM [18].A thorough screening by a Pulmonary physician, experienced in TB management could at least partly explain the higher yield in our study. There has been few attempts to

**Table 2. The socio-demographic and relevant DM treatment and control variables.**

| Characteristics | DM patients with TB disease | DM patients with no TB disease | p-value |
|---|---|---|---|
| Total | 53 | 947 | |
| Sex | | | |
| Male | 38 (71.7) | 538 (56.8) | 0.033 |
| Female | 15 (28.3) | 409 (43.2) | |
| Age | | | |
| ≤45 years | 15 (28.3) | 230 (24.3) | 0.51 |
| 46–60 years | 30 (56.6) | 490 (51.7) | 0.487 |
| ≥61 years | 8 (15.1) | 227 (24.0) | 0.137 |
| Occupation | | | |
| Business | 5 (9.4) | 111 (11.7) | 0.611 |
| Home makers | 9 (17.0) | 348 (36.8) | 0.034 |
| Office job | 17 (32.1) | 206 (21.7) | 0.077 |
| Manual labourer | 15 (28.3) | 145 (15.3) | 0.012 |
| Retired | 7 (13.2) | 137 (14.5) | 0.793 |
| Smoking | 14 (26.4) | 134 (14.2) | 0.015 |
| Alcohol | 12 (22.6) | 82 (8.7) | <0.001 |
| BMI, Kg/m2 | | | |
| Mean±SD | 26 | 25 | 0.363 |
| <18 | 5 (5.7) | 28 (3.0) | 0.273 |
| 18–23 | 22 (41.5) | 245 (25.9) | 0.013 |
| >23 | 28 (52.8) | 674 (71.2) | 0.004 |
| Duration of DM years | | | |
| Mean ± SD | 10 | 7 | 0.161 |
| <1 | 4 (7.6) | 90 (9.5) | 0.645 |
| 5-Jan | 13 (24.5) | 338 (35.7) | 0.096 |
| >5 | 36 (67.9) | 272 (54.8) | 0.062 |
| Treatment details | | | |
| OHA | 14 (26.4) | 487 (51.4) | <0.001 |
| Insulin | 21 (39.6) | 193 (20.4) | <0.001 |
| Both | 18 (34.0) | 267 (28.2) | 0.363 |
| Fasting Glucose, Median(IQR) | 156 (124, 222) | 157 (121, 214) | 0.746 |
| Post Prandial Glucose, Median (IQR) | 257 (205, 344) | 239 (182, 316) | 0.084[†] |
| HbA1c | | | |
| <7 | 10 (18.9) | 243 (25.7) | 0.268 |
| 7–8.9 | 23 (43.4) | 369 (38.9) | 0.514 |
| > = 9 | 20 (37.7) | 335 (35.4) | 0.734 |

[a] Chi-square test was used to assess the association between given factors and two groups

use ultrasound to screen for extra pulmonary tuberculosis in high risk individuals in resource limited settings [19,20].

Rieder [21] has reported that age and sex are strong determinants of TB, with the highest risks being found in elderly people, and this was evident in the study by S. Kumpatla et al. [17] also. Systematic reviews have shown that under nutrition, smoking, diabetes and alcohol consumption are individual risk factors that can double or triple the risk of developing active TB. [22] A large part of the TB burden in India has been attributed to smoking (40%) and DM (15%). [23,24] We also observed that male gender, history of smoking and alcohol

**Table 3. Socio-demographic characteristics of 'LTBI' and 'DM Only' group.**

| Characteristics | | | p-value |
|---|---|---|---|
| | LTBI | No TB | |
| **Total** | **96** | **104** | |
| Sex | | | |
| Male | 69 (71.9) | 61 (58.7) | |
| Female | 27 (28.1) | 43 (41.3) | 0.05 |
| Age | | | |
| ≤45 years | 29 (30.2) | 24 (23.1) | 0.256 |
| 46–60 years | 45 (46.9) | 56 (53.8) | 0.329 |
| ≥61 years | 22 (22.9) | 24 (23.1) | 0.973 |
| Occupation | | | |
| Business | 18 (18.8) | 11 (10.6) | 0.1 |
| Home maker | 20 (20.8) | 34 (32.7) | 0.058 |
| Job | 32 (33.3) | 24 (23.1) | 0.109 |
| Labourer | 16 (16.7) | 14 (13.5) | 0.527 |
| Retired | 10 (10.4) | 21 (20.2) | 0.056 |
| Smoking | 24 (25.0) | 17 (16.4) | 0.13 |
| Alcohol | 15 (15.6) | 8 (7.7) | 0.079 |
| BMI | | | |
| <18 | 2 (2.1) | 7 (6.7) | 0.117 |
| 18–23 | 26 (27.1) | 35 (33.7) | 0.311 |
| >23 | 68 (70.8) | 62 (59.6) | 0.097 |
| Duration | | | |
| <1 year | 6 (6.2) | 11 (10.6) | 0.265 |
| 1–5 years | 31 (32.3) | 31 (29.8) | 0.703 |
| >5 years | 59 (61.5) | 62 (59.6) | 0.784 |
| Treatment | | | |
| OHA | 51 (53.1) | 58 (55.8) | 0.702 |
| Insulin | 19 (19.8) | 19 (19.3) | 0.929 |
| Both | 26 (27.1) | 27 (25.9) | 0.848 |

[a] Chi-square test was used to assess the association between given factors and two groups

consumption were strongly associated with TB diagnosis in Diabetic patients. This makes a stronger case for screening DM patients who have the above additional risk factors.

Compared to no 'DM-TB' patients, 'DM Only' patients had higher body weight [25]. In our study this was true and significantly more 'DM Only' had high BMI.

S. Kumpatlaetal [17], observed that 'DM-TB' group were more likely to have metabolic disease, (>10 years), to be on combination therapy with oral and insulin medication and have higher HB1AC levels (>7). This indicates that DM patients with TB had more severe disease requiring more treatment for control. [26] Our 'DM-TB' patients were more likely to be on insulin rather than oral hypoglycemic agents only. This could imply that 'DM-TB' patients had more severe DM. It is very important that control of diabetes is given due importance in patients with tuberculosis [27].

Tuberculin skin test surveys in India show a very high annual risk of TB infection. Given the high TB burden of active TB in India, it is not surprising that nearly 40% of Indians are estimated to be latently infected. Despite the likelihood of a large number of LTBI reactivating to active TB, the National program (RNTCP) does not give priority to LTBI detection and

treatment in the public sector. This is true for most of the other high TB burden countries around the world. [28]

We have demonstrated a high (48%) prevalence of LTBI in our diabetic patients, which is higher than the prevalence of LTBI reported for the Indian population. We did not observe any socio-demographic risk factor for latent TB infection in diabetic patients, however the population tested was small. With regard to the duration of Diabetes, treatment and glycemic control, there was no difference between 'DM-LTBI' and 'No LTBI' suggesting that other factors may also be responsible for the susceptibility to LTBI. However, Gerardo Martınez-Aguilar et al [29] found a higher proportion of subjects with HbA1c values of more than 7% in the LTBI (TST-positive) than in the TST negative group.

## Limitations

One of the drawbacks of the study is not having done Interferon Gamma Release Assays due to local issues with availability, when the study commenced.

## Conclusion

The study highlights that screening of DM patients for active TB is feasible and productive in DM clinics in referral hospitals. This could lead to increase on case detection as well as earlier diagnosis. Screening should include assessment for extra-pulmonary TB as well.

## Author Contributions

**Conceptualization:** Pradipkumar Arvindbhai Dabhi, Balamugesh Thangakunam, Devasahayam Jesudas Christopher.

**Data curation:** Devasahayam Jesudas Christopher.

**Formal analysis:** Pradipkumar Arvindbhai Dabhi.

**Funding acquisition:** Devasahayam Jesudas Christopher.

**Investigation:** Pradipkumar Arvindbhai Dabhi, Dukhabandhu Naik.

**Methodology:** Pradipkumar Arvindbhai Dabhi, Balamugesh Thangakunam, Richa Gupta, Prince James, Nihal Thomas, Dukhabandhu Naik, Devasahayam Jesudas Christopher.

**Project administration:** Devasahayam Jesudas Christopher.

**Supervision:** Balamugesh Thangakunam, Richa Gupta, Prince James, Nihal Thomas, Devasahayam Jesudas Christopher.

**Validation:** Balamugesh Thangakunam.

**Writing – original draft:** Pradipkumar Arvindbhai Dabhi, Devasahayam Jesudas Christopher.

**Writing – review & editing:** Balamugesh Thangakunam, Richa Gupta, Prince James, Nihal Thomas, Dukhabandhu Naik, Devasahayam Jesudas Christopher.

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
