## [Decision Letter · Decision Letter 0]

6 Apr 2020

PONE-D-20-05049

Screening for Prevalence of Current TB disease and latent TB infection in Type 2 Diabetes Mellitus patients attending a Diabetic clinic in a Indian tertiary care hospital

PLOS ONE

Dear Dr. Christopher,

Thank you for submitting your manuscript to PLOS ONE. After careful consideration, we feel that it has merit but does not fully meet PLOS ONE’s publication criteria as it currently stands. Therefore, we invite you to submit a revised version of the manuscript that addresses the points raised during the review process.

We would appreciate receiving your revised manuscript. To enhance the reproducibility of your results, we recommend that if applicable you deposit your laboratory protocols in protocols.io, where a protocol can be assigned its own identifier (DOI) such that it can be cited independently in the future. For instructions see: http://journals.plos.org/plosone/s/submission-guidelines#loc-laboratory-protocols

We look forward to receiving your revised manuscript.

Kind regards,

Frederick Quinn

Academic Editor

PLOS ONE

Reviewers' comments:

Reviewer's Responses to Questions

**Comments to the Author**

1. Is the manuscript technically sound, and do the data support the conclusions?

Reviewer #1: Yes

Reviewer #2: Partly

Reviewer #3: Yes

2. Has the statistical analysis been performed appropriately and rigorously? 

Reviewer #1: Yes

Reviewer #2: No

Reviewer #3: Yes

3. Have the authors made all data underlying the findings in their manuscript fully available?

Reviewer #1: Yes

Reviewer #2: No

Reviewer #3: Yes

4. Is the manuscript presented in an intelligible fashion and written in standard English?

Reviewer #1: Yes

Reviewer #2: No

Reviewer #3: Yes

5. Review Comments to the Author

Reviewer #1: Authors wrote a very interesting paper, on an important issue.

The link beetwen comunicable (TB) and non communicable diseases is very modern issue.

Well done.

Only some suggestion to make the manuscript clearer and more complete

1. Introduction: explain better the burden of TB in your country and compare it with global burden both TB and DM

and why they are linked (ex Pizzol D, Tuberculosis and diabetes: current state and future perspectives. Trop Med Int Health. 2016 Jun;21(6):694-702)

2. Methods: well wrote

3. Results: 1000 patients are a very important ssample. Well done.

4. Discussion : If you can compare better with:

4a data from other low income setting (ex Pizzol D et al. Prevalence of diabetes mellitus in newly diagnosed pulmonary tuberculosis in Beira, Mozambique.Afr Health Sci. 2017 Sep;17(3):773-779.);

4b. how is important treat and not only test diabetes (Di Gennaro f et al Diabetes in active tuberculosis in low-income countries: to test or to take care?Lancet Glob Health. 2019 Jun;7(6):e707.

4c. How diabetes is a predictors failure of tb(ex Pizzol D et alPredictors of therapy failure in newly diagnosed pulmonary tuberculosis cases in Beira, Mozambique.BMC Res Notes. 2018 Feb 5;11(1):99)

4d. and how frugal tecnology as ultrasound can help tb diagnosis in poor setting (Bobbio F et al Focused ultrasound to diagnose HIV-associated tuberculosis (FASH) in the extremely resource-limited setting of South Sudan: a cross-sectional study.BMJ Open. 2019 Apr 2;9(4):e027179.and Di Gennaro F et al Potential Diagnostic Properties of Chest Ultrasound in Thoracic Tuberculosis-A Systematic Review.Int J Environ Res Public Health. 2018 Oct 12;15(10)

Increase your references and explain better this correlation but the paper is already well written

Reviewer #2: This manuscript has no innovation, just simplely collected and canculated the data. The authors need to described and analyzed the data in manuscript in more details. I confirmed that it not suitable for publication in "PLoS One". I have no comments to the authors.

Reviewer #3: In the manuscript titled “Screening for Prevalence of Current TB disease and latent TB infection in Type 2 Diabetes Mellitus patients attending a Diabetic clinic in a Indian tertiary care hospital”, the authors investigated the prevalence of TB disease and latent TB infection in Type 2 diabetes patients and identified several risk factors for TB diseases. The study is generally well designed and should be of interest to the field. Should my concerns below be properly addressed, I will support the publication of the manuscript in Plos One journal. Please see below for comments.

Line 182 It should be “4” rather than “3” categories

Line 207 Have you got the BCG vaccination history of those patients? If a patient was BCG vaccinated, a positive TST test may not mean he/she has LBTI. A Quantiferon blood test should be considered to confirm it is a real LBTI.

Line 250 It should be Table 2 not Table 3.

Line 250-251 What is normal BMI? The authors should give a definition. It seems that the authors considered BMI>23 as normal, please justify it by providing references.

Line 254-255 It says, “There was no difference between the groups with regards to HbA1c, fasting and post prandial plasma glucose”. I cannot find data of fasting and post prandial plasma glucose in Table2, please include.

Line 258 should be “59%” not “9%”

Line 260 delete “DM”

6. PLOS authors have the option to publish the peer review history of their article (what does this mean?). If published, this will include your full peer review and any attached files.

Reviewer #1: Yes: Francesco Di Gennaro

Reviewer #2: No

Reviewer #3: No

---

## [Author Response · Author response to Decision Letter 0]

29 Apr 2020

We have addressed all the queries and revised the "Manuscript" as per reviewers comments. Appropriate references have been included in the revised manuscript.

---

## [Decision Letter · Decision Letter 1]

5 May 2020

Screening for Prevalence of Current TB disease and latent TB infection in Type 2 Diabetes Mellitus patients attending a Diabetic clinic in a Indian tertiary care hospital

PONE-D-20-05049R1

Dear Dr. Jesudas Christopher,

We are pleased to inform you that your manuscript has been judged scientifically suitable for publication and will be formally accepted for publication once it complies with all outstanding technical requirements.

With kind regards,

Frederick Quinn

Academic Editor

PLOS ONE

Additional Editor Comments (optional):

Reviewers' comments:

Reviewer's Responses to Questions

**Comments to the Author**

1. If the authors have adequately addressed your comments raised in a previous round of review and you feel that this manuscript is now acceptable for publication, you may indicate that here to bypass the “Comments to the Author” section, enter your conflict of interest statement in the “Confidential to Editor” section, and submit your "Accept" recommendation.

Reviewer #1: All comments have been addressed

Reviewer #2: All comments have been addressed

Reviewer #3: All comments have been addressed

2. Is the manuscript technically sound, and do the data support the conclusions?

Reviewer #1: Yes

Reviewer #2: Yes

Reviewer #3: Yes

3. Has the statistical analysis been performed appropriately and rigorously? 

Reviewer #1: Yes

Reviewer #2: Yes

Reviewer #3: Yes

4. Have the authors made all data underlying the findings in their manuscript fully available?

Reviewer #1: Yes

Reviewer #2: Yes

Reviewer #3: Yes

5. Is the manuscript presented in an intelligible fashion and written in standard English?

Reviewer #1: Yes

Reviewer #2: (No Response)

Reviewer #3: Yes

6. Review Comments to the Author

Reviewer #1: Authors improve their manuscript, I really appreciate the manuscript

Research on the link between communicable and noncommunicable diseases is very modern.

I suggest to pubblish it

Reviewer #2: The revised manuscript covered all the comments, and is suitable for publication now. I have no comments to authors.

Reviewer #3: The authors have revised the manuscript according to my comments, therefore I support its publication in this journal.

7. PLOS authors have the option to publish the peer review history of their article (what does this mean?). If published, this will include your full peer review and any attached files.

Reviewer #1: Yes: Francesco Di Gennaro

Reviewer #2: No

Reviewer #3: No

---

## [Editor Report · Acceptance letter]

27 May 2020

PONE-D-20-05049R1 

Screening for Prevalence of Current TB disease and latent TB infection in Type 2 Diabetes Mellitus patients attending a Diabetic clinic in a Indian tertiary care hospital 

Dear Dr. Christopher:

I am pleased to inform you that your manuscript has been deemed suitable for publication in PLOS ONE. Congratulations! Your manuscript is now with our production department. 

With kind regards,

on behalf of

Dr. Frederick Quinn 

Academic Editor

PLOS ONE